# Better Correlation and Robustness: A Distribution-Balanced Self-Supervised Learning Framework for Automatic Dialogue Evaluation

**Peiwen Yuan, Xinglin Wang, Jiayi Shi, Bin Sun, Yiwei Li, Kan Li**[*]
School of Computer Science and Technology, Beijing Institute of Technology, China
{peiwenyuan,wangxinglin,shijiayi,binsun,liyiwei,likan}@bit.edu.cn

## Abstract

Turn-level dialogue evaluation models (TDEMs), using self-supervised learning (SSL) framework, have achieved state-of-the-art performance in open-domain dialogue evaluation. However, these models inevitably face two potential problems. First, they have low correlations with humans on medium coherence samples as the SSL framework often brings training data with unbalanced coherence distribution. Second, the SSL framework leads TDEM to nonuniform score distribution. There is a danger that the nonuniform score distribution will weaken the robustness of TDEM through our theoretical analysis. To tackle these problems, we propose **B**etter **C**orrelation and **R**obustness (BCR), a distribution-balanced self-supervised learning framework for TDEM. Given a dialogue dataset, BCR offers an effective training set reconstructing method to provide coherence-balanced training signals and further facilitate balanced evaluating abilities of TDEM. To get a uniform score distribution, a novel loss function is proposed, which can adjust adaptively according to the uniformity of score distribution estimated by kernel density estimation. Comprehensive experiments on 17 benchmark datasets show that vanilla BERT-base using BCR outperforms SOTA methods significantly by 11.3% on average. BCR also demonstrates strong generalization ability as it can lead multiple SOTA methods to attain better correlation and robustness. Code and datasets: https://github.com/ypw0102/Better-Correlation-and-Robustness.

## 1 Introduction

Evaluating model-generated responses efficiently and accurately can facilitate the hyperparameter tuning and comparison among models, which is essential for the research of open-domain dialogue system (Bao et al., 2020; Sun et al., 2021; Feng et al., 2021; Li et al., 2023). Therefore, economical and practical automatic metrics are widely applied instead of human evaluation during development phase. However, the referenced automatic metrics, assessing dialogue based on the golden response (e.g. BLEU (Papineni et al., 2002) and BERTScore (Zhang et al., 2020b)), have been shown to be inaccurate for dialogue evaluation (Liu et al., 2016; Deriu et al., 2021) due to the one-to-many nature of dialogue (Zhao et al., 2017). Fortunately, the unreferenced metrics, especially the turn-level dialogue evaluation models (TDEMs) (Mehri and Eskénazi, 2020b,a; Huang et al., 2020; Ye et al., 2021; Zhang et al., 2021a, 2022b), have achieved a favorable budget between efficiency and accuracy.

Nevertheless, existing TDEMs still perform worse correlations with human judgements on medium coherence samples (samples with human ranking scores falling within the interval [0.25, 0.75]) and lack robustness as shown in Figure 1. We argue that these problems stem from the self-supervised learning (SSL) paradigm, which generally constructs positive and negative samples to strengthen

---

[*]Corresponding author.

37th Conference on Neural Information Processing Systems (NeurIPS 2023).

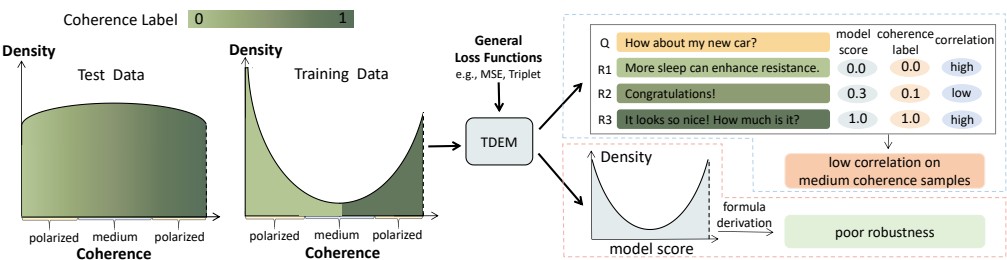

Figure 1: Deficiencies of self-supervised TDEM. (a) TDEM possesses low correlations with human judgements on conversations of medium coherence due to the lack of such samples of training data. (b) TDEM attains poor robustness due to the nonuniform score distribution caused by discrete label domain of training data together with general fixed loss functions.

the scoring ability of TDEM. Due to the fact that positive samples possess high coherence while negative samples possess low coherence (Sato et al., 2020), TDEM lacks training signals of medium coherence samples, thus leading to a bad performance when evaluating on such samples (e.g., (Q, R2) in Figure 1). Moreover, current discrete label domain together with fixed loss functions generally brings TDEM a nonuniform score distribution (Figure 1, 3), which will hurt the robustness of TDEM according to our theoretical analysis (demonstrated in §3.4).

To solve the aforementioned problems, we propose **B**etter **C**orrelation and **R**obustness (BCR), a distribution-balanced SSL framework for TDEM. BCR offers a **T**raining **S**et **R**econstructing (TSR) method to balance the data coherence distribution of training set and advance a high correlation with humans on medium coherence samples, and a **D**ynamic **P**enalty loss function (DP loss) for a more uniform model score distribution. In TSR method, we use two rule-based strategies to expand a conversation into a positive sample and a negative sample respectively to supplement medium coherence samples. Considering that utterances within a conversation are internally coherent, our strategies are reasonable. We also provide continuous labels for these medium coherence samples through self-distillation to alleviate the deviation of discrete two-level labels. DP loss promotes TDEM to obtain a more uniform score distribution by adaptively adjusting the penalty strength of samples in different scoring intervals according to the uniformity of score distribution estimated by kernel density estimation (Parzen, 1962), thus enhancing the robustness. Our contributions are summarized as follows:

- We propose Better Correlation and Robustness (BCR), a novel distribution-balanced SSL framework for TDEM, which can effectively alleviate limitations of vanilla SSL framework through Training Set Reconstructing (TSR) and Dynamic Penalty loss function (DP loss).
- We theoretically prove that the robustness of TDEM and the uniformity of score distribution are positively correlated, and further verify it through experiments.
- Comprehensive experiments on 17 benchmark datasets show that vanilla BERT-base (Devlin et al., 2019) applying BCR can outperform SOTA methods significantly by 11.3% on average. We also demonstrate the strong generalization of BCR as it can lead multiple SOTA methods to attain better robustness and correlations with human judgements.

## 2   Related work

Referenced metrics have shown to correlate poorly with human judgements (Liu et al., 2016) and thus various unreferenced model-based metrics have been proposed, which can be divided into CDEM (conversation-level) and TDEM (turn-level). CDEMs (Mehri and Eskénazi, 2020a; Ghazarian et al., 2022; Zhang et al., 2022a) evaluate conversations between real user and dialogue system. As this interactive evaluation occupies too much extra energy of researchers, we mainly study TDEM, which can be divided into supervised and self-supervised.

Supervised TDEMs (Lowe et al., 2017; Ye et al., 2021) are currently hard to train due to the lack of human annotated datasets. It is worth mentioning that the reward models used in the RLHF stage of LLM training (e.g., InstructGPT (Ouyang et al., 2022)) are essentially supervised TDEMs. Although existing literature has not published the performance of these models, we proved through experiments

that as a pre-training method, BCR can help supervised TDEMs converge to better performance faster (see Appendix C.2). This may inspire scholars to research and apply pre-trained TDEMs to improve the performance of reward models, thus attaining better LLMs.

Self-supervised TDEMs generally apply next utterance prediction as training task, in which the key points are training set construction and training loss design. Original response and randomly sampled response are generally regarded as positive and negative samples respectively for a given context (Tao et al., 2018; Phy et al., 2020). Some literature attains hard negative samples through Word2Vec similarity based filtering (Zhang et al., 2022b) , speaker-level utterance shuffling (Zhang et al., 2021a) , model generation (Lan et al., 2020) and adversarial craft (Sai et al., 2020; Ye et al., 2021). However, the lack of medium coherence samples will lead TDEMs to unbalanced evaluating ability. General loss functions are widely used to train TDEMs, such as Triplet loss (Tao et al., 2018), CrossEntropy loss (Mehri and Eskénazi, 2020b), NCE loss (Sinha et al., 2020), etc. (Ye et al., 2021) applies three loss functions and attains a score distribution with three peaks. All of these loss functions together with discrete label domain generally lead to a nonuniform score distribution, which will hurt the robustness of TDEMs, as proven in 3.4.

## 3 Preliminary

In this section, we first introduce the task definition and a baseline model, followed by verifying the low correlations and poor robustness caused by the SSL framework.

### 3.1 Task Definition

Given a conversation triplet (*context, query, response*) where *response* is offered by a dialogue system, TDEM is required to measure the quality (e.g., coherence) of the *response*. We can calculate the correlation coefficient (e.g., Spearman Correlation) with human judgements on annotated datasets to evaluate the performance of TDEM.

### 3.2 Baseline Model

Our baseline model is shown in Figure 2. Specifically, it contains a BERT-base-uncased model (Devlin et al., 2019) as feature extractor and a three-layers multi-layer perceptron to get a coherence score distributed within interval [0, 1]. Apart from general embeddings (token, speaker, position), we additionally apply class embeddings to distinguish utterances from *context*, *query* and *response* as the coherence between *query* and *response* is the dominant judging factor (Li et al., 2021). Following GRADE (Huang et al., 2020), we use Triplet loss to train the baseline model on the DailyDialog[2] dataset (Li et al., 2017). The gap of Triplet loss is set as 0.3 at which our baseline model can attain best performance. For each conversation, we randomly select T (T>1) consecutive turns as positive sample and replace the last utterance with a randomly chosen utterance to get negative sample.

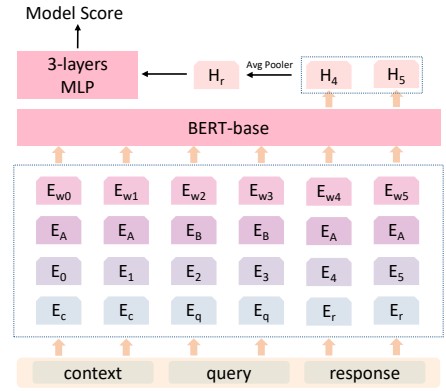

Figure 2: Baseline model. We apply four kinds of embeddings from top to bottom: token embedding, speaker embedding, position embedding and class embedding.

### 3.3 Coherence Distribution

To verify the impact of unbalanced coherence distribution of training data, we test our baseline model on subsets of DSTC10 Zhang et al. (2021b) dataset with different coherence respectively (Table 2). Our baseline model (w/o. TSR) obtains 0.125 Spearman correlation at the polarized interval while only 0.073 at the medium interval. This proves that TDEM applying general SSL framework performs bad on medium coherence samples.

---

[2]http://yanran.li/dailydialog

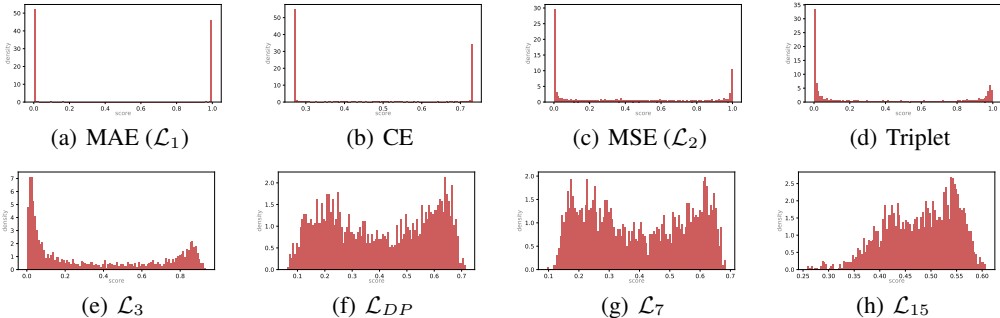

Figure 3: Score distribution of TDEM trained with different loss functions on DSTC10 dataset.

### 3.4 Score Distribution

As shown in Figure 3, widely used loss functions (MAE, MSE, Triplet, CrossEntropy) lead TDEM to a polarized score distribution under the discrete label domain setting. We conjecture that these loss functions continuously give a relatively significant penalty loss to the samples already scoring within polarized interval, which aggravate this polarized trend.

We primarily consider the influence of score distribution on the robustness of TDEM. Given a small disturbance $\epsilon$ at the sample, the corresponding predicted score changes $\lambda$, which will affect Spearman correlations $r_s$ between the original score $x$ and the new score $x + \lambda$. Robust TDEM is supposed to resist noise, thus attains higher $r_s$. Hence, we can use $\mathbb{E}(r_s)$, the mathematical expectation of $r_s$ on $n$ samples, to reflect the robustness of TDEM, and make the following statement:

**Theorem 1** *For any $f(x)$, the probability density function of TDEM score distribution, $\mathbb{E}(r_s)$ has an upper bound after a small disturbance:*

$$\mathbb{E}(r_s) \leq 1 - \frac{6\mathbb{E}(\lambda)^2}{n^2 - 1}, \tag{1}$$

*and the equality condition is $f(x) \equiv 1, \forall x \in [0, 1]$.*

**Proof 1** *The ranking difference $d(x)$ before and after disturbance is :*

$$d(x) = \int_x^{x+\lambda} f(x)dx \tag{2}$$

*According to the definition of Spearman correlations, $E(r_s)$ can be written as:*

$$\mathbb{E}(r_s) = \mathbb{E}(1 - \frac{6 \sum_{i=1}^n d(x_i)^2}{n(n^2 - 1)}), \tag{3}$$

*we derive the lower bound of $\mathbb{E}(d(x)^2)$ as follows (See detailed derivation in Appendix A):*

$$\begin{aligned} \mathbb{E}(d(x)^2) &= \int_0^1 \left( \int_x^{x+\mathbb{E}(\lambda)} f(u)du \right)^2 f(x)dx \\ &\geq \left( \int_0^1 \int_x^{x+\mathbb{E}(\lambda)} f(u)duf(x)dx \right)^2 \\ &\geq \mathbb{E}(\lambda)^2 \end{aligned} \tag{4}$$

*The equality condition of Eq. (4) is $f(x) \equiv 1$ for $x \in [0, 1]$. Taking the lower bound of $\mathbb{E}(d(x)^2)$ into Eq. (3), we conclude the proof.*

Note that higher $\mathbb{E}(r_s)$ denotes better robustness of TDEM. Hence, we can derive that the robustness of TDEM correlates positively with the uniformity of score distribution based on Theorem 1.

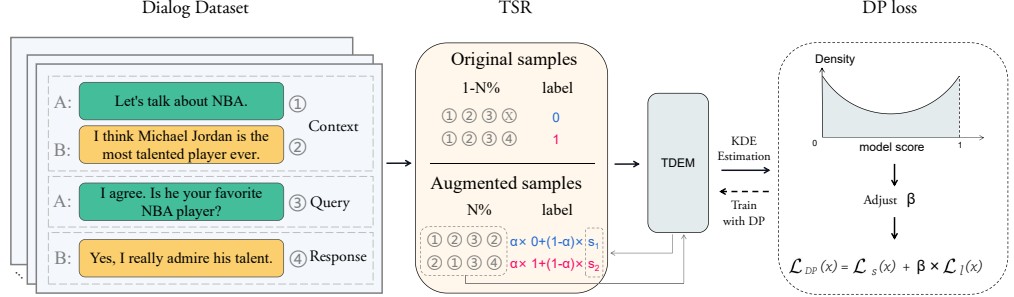

Figure 4: Illustration of BCR Framework.

# 4 BCR framework

We propose BCR framework (Figure 4) to alleviate the low correlations on medium coherence samples and poor robustness caused by SSL framework. BCR offers: (1) *training set reconstructing* (TSR) to attain coherence-balanced training samples with continuous label domain (§4.1) ; (2) DP loss to bring TDEM a more uniform score distribution and better robustness (§4.2).

## 4.1 Balance Coherence Distribution

As training set of SSL framework generally lacks samples of medium coherence, we propose TSR to reconstruct a coherence balanced training set. Specifically, We replace $N\%$ samples of the original dataset $\mathbb{D}_{ori}$ with samples of medium coherence generated through two strategies, denoted as $\mathbb{D}_{aug}$.

Given a conversation, we suppose that replacing response with an utterance of context generally leads to a negative sample with medium coherence and disrupting context while maintaining the query and response leads to a positive sample with medium coherence. Our strategies are motivated by (Zhang et al., 2021a), which shuffles all utterances of a speaker to attain a hard negative sample. As a comparison, the samples generated by our strategies are more controllable in terms of coherence. Take Figure 4 for example. As we randomly replace ④ with ②, although ② does not answer ③, they have the same topic (*NBA player*). Thus, (①, ②, ③, ②) constitute a negative sample with medium coherence. As we exchange the order of ① and ②, it remains a positive example since ④ still answers ③. However, the word *talent* in ④ seems unnatural as ② is no longer said by Speaker B, which makes (②, ①, ③, ④) a positive sample with medium coherence. Further human evaluation experiments show that TSR can provide stronger training signals with medium coherence compared with advanced data augmentation methods (Appendix B.4).

Assigning discrete two-level labels to samples with medium coherence can result in serious deviation. Therefore, we combine the original discrete labels and scores of TDEM to attain soft pseudo labels for the augmented samples and the final labels of the whole reconstructed dataset are as follows:

$$Label_{TSR}(x) = \begin{cases} Label_{discrete}(x), \ x \in \mathbb{D}_{ori} \\ \alpha \times Label_{discrete}(x) + (1-\alpha) \times \mathcal{M}(x), \ x \in \mathbb{D}_{aug} \end{cases} \quad (5)$$

where $\mathcal{M}$ denotes TDEM, $Label_{discrete}(x) = 1$ if $x$ is a positive sample and 0 otherwise. During training process, pseudo labels are obtained in real time.

## 4.2 Balance Score Distribution

Based on Theorem 1, we consider adaptively approaching the optimal loss function for a uniform score distribution to strengthen the robustness of TDEM. According to Weierstrass Approximation Theorem (Weierstrass, 1885) , we consider approximating the optimal loss function in with polynomial loss function $\mathcal{L}(x)$:

$$\mathcal{L}_i(x) = \mid \mathcal{M}(x) - Label_{TSR}(x) \mid^i \quad (6)$$

$$\mathcal{L}(x) = \sum_{i=0}^{\infty} \beta_i \mathcal{L}_i(x) \quad (7)$$

To gain a deeper insight of the relationship between the probability density function of TDEM score distribution $f(x)$ and $\mathcal{L}_i(x)$ , we propose Loss Ratio Curve (Lrc) as a monitor. Taking the difference value (D-value) between label and predicted score as X-axis, the ratio of the corresponding loss and fiducial loss (loss when D-value is 0.5) as Y-axis, we get Lrc shown in Figure 5. Lrc visualizes the penalty strength (loss ratio) at different D-value of a certain loss function. The stronger concavity Lrc possesses, the smaller penalty the samples with polarized scores attain. Combining Figure 3 and Figure 5, score distribution undergoes a transition from polarization to centralization with the increasing concavity of Lrc and the rising power of $\mathcal{L}_i(x)$. This not only confirms our conjecture in Section 3.4, but also indicates that

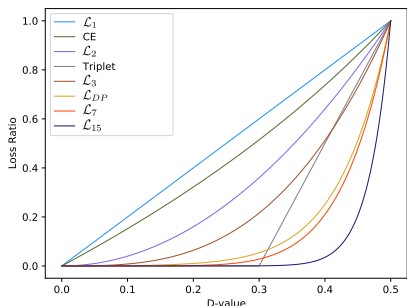

Figure 5: Loss ratio curve of different loss functions

$\mathcal{L}_i(x)$ with small power (e.g., i = 3) generally leads TDEM to a polarized distribution while $\mathcal{L}_i(x)$ with large power (e.g., i = 15) promotes a centralized distribution.

Bared this insight in mind, we can simplify Eq. (7) and attain our DP loss function as follows:

$$\mathcal{L}_{DP}(x) = \mathcal{L}_s(x) + \beta\mathcal{L}_l(x) \tag{8}$$

where $s$ is smaller than $l$. During training, $\mathcal{L}_s(x)$ and $\mathcal{L}_l(x)$ promote polarized and centralized score distribution respectively, and $\beta$ adjusts adaptively according to the score distribution to make TDEM converge stably to a more uniform distribution. Specifically, we use kernel density estimation (KDE) to estimate the score distribution of TDEM on training set $\mathbb{D}$ at the end of each epoch as follows:

$$\hat{f}(x) = \frac{1}{|\mathbb{D}|h} \sum_{i=1}^{|\mathbb{D}|} K\left(\frac{x - X_i}{h}\right) \tag{9}$$

where $|\mathbb{D}|$ is the size of the training set, $h$ is the bandwidth of KDE, $X_i$ is the model score of the $i^{th}$ sample and $K(\cdot)$ is the kernel function. We divide [0, 1] into polarized interval ([0, 0.25] and [0.75, 1]) and centralized interval ([0.25, 0.75]). If the estimated distribution shows a polarized trend (the integral of $\hat{f}(x)$ in polarized interval > 0.6), we update $\beta_{new} = \beta_{old} \times 10$ to enhance the influence of $\mathcal{L}_l$, so as to alleviate such trend. On the contrary, we update $\beta_{new} = \beta_{old} / 10$. From a more intuitive perspective, DP loss can dynamically adjust the penalty strength on different samples to approximate the optimal loss function, thereby enabling TDEM to achieve a relatively uniform score distribution.

# 5  Experiments

We first apply BCR framework on BERT-base, and compare it with the SOTA methods on multiple datasets to verify its effectiveness. Then, we apply BCR framework on existing SOTA TDEMs to verify the generalization of BCR and whether it can lead TDEMs to attain better robustness and correlations with human judgements. Ablation study and case study are followed to further understand the effectiveness of BCR. [3]

## 5.1  BERT-base With BCR

### 5.1.1  Experimental Setup

We apply BCR framework to train the baseline model mentioned in Section 3.2 on DailyDialog dataset. We set learning rate as 5e-5, batch size as 32, $\alpha$ in Eq. (5) as 0.8, initial value of $\beta$ as 1, $N$ in TSR as 20. The default value of $s$ and $l$ in Eq. (8) are 3 and 7, and we examined the effects of different values in 5.3. In Eq. (9), we use Gaussian kernel and set h as 0.01. AdamW Loshchilov and Hutter (2017) is used as the optimizer (see Appendix B.2 for details).

We conduct experiments on 17 benchmark quality-annotated datasets and compare the results with SOTA methods of each dataset (see Table 5). As DSTC10 datasets contain five sub-datasets and

---

[3]For each experiment, We run four random seeds and report the averaged result.

Table 1: Spearman correlations between TDEMs and human judgements on 17 datasets, with standard deviations in gray. BERT refers to baseline model (See Section 3.2). The results of SOTA TDEMs come from the corresponding papers. All results are statistically significant (p-value < 0.05).

| Method | FED | GCG | GCR | GDG | GDR | GEG | GER | PE | DSTC6 |
|---|---|---|---|---|---|---|---|---|---|
| BERT | 0.205 | 0.606 | 0.500 | 0.310 | 0.197 | 0.202 | 0.315 | 0.612 | 0.224 |
| SOTA | 0.264 | 0.617 | **0.558** | 0.358 | 0.187 | 0.223 | 0.338 | **0.699** | **0.295** |
| BERT+BCR | **0.286** | **0.642** | 0.519 | **0.416** | **0.487** | **0.341** | **0.372** | 0.669 | 0.274 |

| Method | UT | UP | JSALT | ESL | NCM | DT | DP | DSTC10 | **Average** |
|---|---|---|---|---|---|---|---|---|---|
| BERT | 0.305 | 0.372 | 0.166 | 0.345 | 0.253 | 0.220 | 0.401 | 0.255 | $0.323 \pm 0.012$ |
| SOTA | 0.419 | **0.469** | 0.116 | 0.414 | 0.299 | **0.326** | **0.456** | 0.310 | 0.373 |
| BERT+BCR | **0.421** | 0.435 | **0.272** | **0.531** | **0.336** | 0.305 | 0.428 | **0.316** | **0.415** $\pm 0.005$ |

corresponding 11 qualities, apart from the appropriateness quality shared by the five sub-datasets, we also report the average results (column DSTC10 in Table 1) of all the 11 qualities. Results of SOTA methods come from the corresponding papers. We also reproduced the results of GRADE (Huang et al., 2020) and USR (Mehri and Eskénazi, 2020b) for a convincing comparison (Figure 7).

### 5.1.2 Experimental Results

As shown in Table 1, BERT-base applying our BCR framework obtains the highest correlations with human judgements on 11 of 17 datasets and 11.3% (4.2 points) higher average Spearman correlations than SOTA methods, which demonstrates the effectiveness of BCR framework (see Appendix B.3 for the results of Pearson correlations). We further visualize the encoded features and the predicted scores through Principal Component Analysis (PCA). As shown in Figure 6, the transition range (marked with red box) from high score to low score is very narrow for baseline model while becomes much broader when applying BCR. This feature distribution can bring two benefits: (a) more discriminative features bring more accurate judgments; (b) larger feature space can better resist noise. This sheds light on the reason for the better robustness and correlations brought by BCR from a deeper perspective.

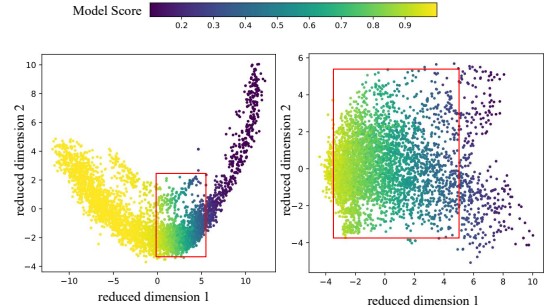

Figure 6: PCA results of baseline model without (left) and with (right) BCR on DSTC10 datasets.

### 5.2 SOTA TDEMs With BCR

### 5.2.1 Experimental Setup

We apply BCR on the following two TDEMs that attain best performance in research (Yeh et al., 2021) and test them on DSTC10 datasets.

- **GRADE** (Huang et al., 2020) possesses a BERT branch to get utterance-level contextualized representations and a Graph-Reasoning branch to get topic-level graph representations.
- **USR** (Mehri and Eskénazi, 2020b) employs one mask language model and two retrieval models to measure sub-qualities of given sample and combines them into an overall score.

We replace the original loss with DP loss and apply TSR based on their own training set for both of the two models to apply BCR. DailyDialog dataset is used to train all the compared TDEMs for a fair comparison. We also measured the difference in Spearman correlations when testing TDEMs on the DSTC10 datasets with and without noise to evaluate the robustness:

$$Diff(\mathcal{M}) = Spearman(\mathcal{M}, AddNoise(\mathbb{D}_{test})) - Spearman(\mathcal{M}, \mathbb{D}_{test}) \qquad (10)$$

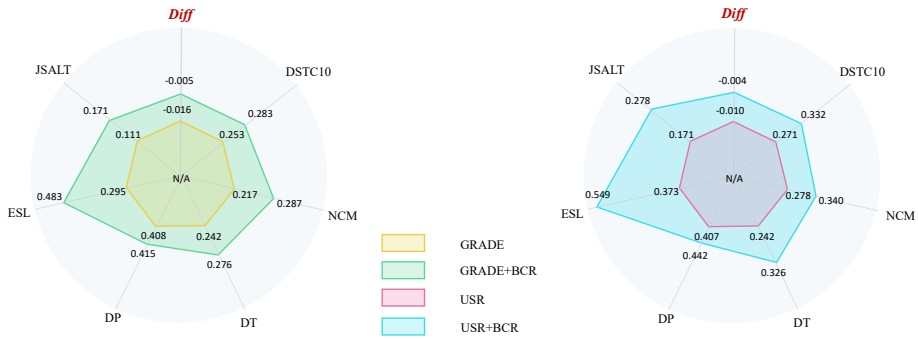

Figure 7: $Diff$ and Spearman correlations of SOTA TDEMs on DSTC10 datasets.

where $\mathbb{D}_{test}$ denotes test dataset, $\mathcal{M}$ denotes TDEM and we randomly drop 15% of words in context and replace 10% of words in both context and query with synonyms to add noise. [4]

### 5.2.2 Experimental Results

As shown in Figure 7, both GRADE and USR attain higher Spearman correlations and $Diff$ when applying BCR framework, which verifies that BCR can lead SOTA TDEMs to attain better robustness and correlations with human judgements. This also confirms the stable generalization of BCR framework.

### 5.3 Ablation Study

We perform ablation studies for the main components of BCR to better analyze their relative contributions. We conduct experiments on the DSTC10 datasets based on BCR+BERT.

**Training Set Reconstructing.** We first verify the effectiveness of TSR. As shown in Table 2, BERT+BCR drops 0.006 and 0.055 Spearman correlations without pseudo label and the whole TSR respectively. From a more fine-grained perspective, baseline model achieves 79.4% (0.058) Spearman correlations gain on medium coherence samples and 30.4% (0.038) Spearman correlations gain on polarized coherence samples respectively when applying TSR. This further proves that TSR can simultaneously strengthen the evaluating ability of TDEM on both medium and polarized coherence samples while the former benefits much more.

Table 2: Ablation results (Spearman correlations) of TSR on subsets with different coherence of DSTC10 datasets, with standard deviations in gray. The Medium indicates samples with coherence labels ranking within interval [0.25,0.75] while the Polarized indicates the rest.

| Metrics | Polarized | Medium | Overall |
|---|---|---|---|
| BERT+BCR | $0.163 \pm 0.003$ | $0.131 \pm 0.002$ | $0.316 \pm 0.002$ |
| w/o. Pseudo label | $0.161 \pm 0.002$ | $0.120 \pm 0.001$ | $0.310 \pm 0.002$ |
| w/o. TSR | $0.125 \pm 0.003$ | $0.073 \pm 0.004$ | $0.261 \pm 0.003$ |

**DP Loss Function.** To verify the effectiveness of DP loss and the impact of different values of $s$ and $l$ in Eq. (8), we apply different loss functions to train TDEM respectively. As shown in Figure 8, DP loss generally brings better robustness and Spearman correlations compared with other loss functions when applying different $(s, l)$ couples. We use $Uniformity = -\lg(\sigma)$ to present the uniformity of score distribution, where $\sigma$ is the variance of score distribution. We find that the more uniform the score distribution (greater $Uniformity$) is, the better robustness TDEM (greater $Diff$) attains. This verifies the theory we proved in §3.4 from an experimental perspective.

---

[4]We use EDA tools nlpaug https://github.com/makcedward/nlpaug to add noise.

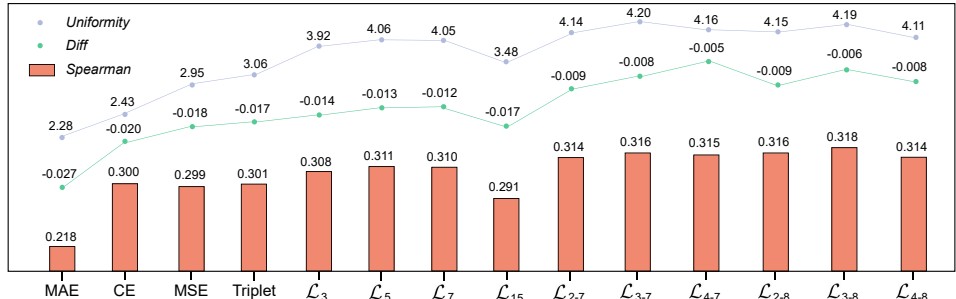

Figure 8: Ablation results of loss functions. $\mathcal{L}_{s-l}$ denotes DP loss applying couple $(s, l)$.

Table 4: Comparison of BCR with respect to parameter count, training costs, and computational costs to the compared methods.

| Method | BCR+BERT | USR | GRADE | MME-CRS |
|---|---|---|---|---|
| Parameter Counts (M) | 110 | 373.8 | 469 | >435.2 |
| Training Costs (hours) | 2 | 6 | 7 | >4 |
| Computational Costs | $T_{base}$ | $3 * T_{base}$ | $T_{base} + T_{GNN}$ | $5 * T_{base}$ |

## 5.4 Case Study

To illustrate the performance of BCR, two representative examples are shown in Table 3. In the first example, the ranking (cumulative distribution function value of score distribution) given by USR and USR+BCR both correlates well with human before adding noise. In case of noise, human judgement stays the same since the meaning of response remains unchanged. However, the ranking given by USR changes sharply (0.83 to 0.99) even though the score barely changes (0.995 to 0.999). USR+BCR resists the noise well by converting the polarized distribution to a more uniform distribution, which verifies the better robustness brought by BCR.

Table 3: Two representative examples show the strength of BCR framework. U1 and U2 are two utterances of the conversation history and R is the corresponding response.

---

**U1**: Did you look in the mirror?
**R**: yeah i did.
Score (Human / USR / USR+BCR): 0.444 / 0.995 / 0.744
Ranking (Human / USR / USR+BCR): 0.81 / 0.83 / 0.83
**R (add noise)**: yes i did.
Score (USR / USR+BCR): 0.999 / 0.732
Ranking (USR / USR+BCR): 0.99 / 0.81

---

**U1**: Why aren't you eating anything else?
**U2**: Well , fruits and vegetables are very healthy.
**R**: What kind of vegetables do you want to eat?
Score (Human / USR / USR+BCR): 0.620 / 0.998 / 0.556
Ranking (Human / USR / USR+BCR): 0.45 / 0.87 / 0.47

---

The second example shows BCR can bring better correlations with human judgements on medium coherence samples.

## 5.5 Efficiency Analysis

Table 4 shows the comparison of BCR with respect to parameter count, training costs, and computational costs to the methods we have examined. Specifically, $T_{base}$ denotes the computational cost for a single pretrained model and $T_{GNN}$ denotes a typical graph neural network. In the three dimensions of comparison, BCR used fewer resources but still achieved better results.

## 6 Conclusion

This paper proposes BCR, a distribution-balanced SSL framework for automatic dialogue evaluation. BCR offers two novel technologies: TSR and DP loss. TSR reconstructs a coherence distribution

balanced training set with continuous label domain. DP loss adjusts adaptively according to the score distribution estimated by kernel density estimation to bring TDEM a more uniform distribution. We prove that the uniformity of score distribution and the robustness of TDEM are positively correlated, which guarantees the better robustness brought by DP loss. Empirical results show that BCR framework brings significant improvements on correlations and robustness to various TDEMs. For future work, we will analyse the pre-training strategies of TDEM to promote the development of dialogue system and LLM training.

**Limitation.** We notice that different TDEMs may produce score disturbance in different scale for a certain small disturbance at the input, which will affect $\mathbb{E}(\lambda)$ in Section 3.4. Fortunately, this does not affect the conclusion of Theorem 1, and we find that for different models with the same backbone network, $\mathbb{E}(\lambda)$ barely changes. We suppose that this is alleviated by the regularization effect of weight decay. We also notice that though DP loss can approximate the optimal loss function, a completely uniform distribution of model score on the test set has not yet been achieved, which we believe is due to the distribution difference in coherence between the training set and the test set. We will investigate how to better align the distribution of training and test sets in the future.

## Acknowledgments and Disclosure of Funding

This work is supported by Beijing Natural Science Foundation (No. 4222037, L181010). We thank the anonymous reviewers for their constructive comments.

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

# A Formula Derivation of Theorem 1

Given

$$\int_0^1 f(x)dx = 1,$$

we perform formula derivation as follows:

$$\int_0^1 \left( \int_x^{x+\mathbb{E}(\lambda)} f(u)du \right)^2 f(x)dx$$

$$= \int_0^1 \left( \int_x^{x+\mathbb{E}(\lambda)} f(u)du \sqrt{f(x)} \right)^2 dx$$

$$= \int_0^1 \left( \int_x^{x+\mathbb{E}(\lambda)} f(u)du \sqrt{f(x)} \right)^2 dx$$

$$\cdot \int_0^1 f(x)dx$$

$$\geq \left( \int_0^1 \int_x^{x+\mathbb{E}(\lambda)} f(u)du f(x)dx \right)^2$$

$$(Cauchy's\ Inequality)$$

$$= \left( \int_0^1 \mathbb{E}(\lambda) \cdot f(x) \cdot f(x)dx \right)^2 \quad (\mathbb{E}(\lambda) \to 0)$$

$$= \mathbb{E}(\lambda)^2 \left( \int_0^1 f(x) \cdot f(x)dx \right)^2$$

$$= \mathbb{E}(\lambda)^2 \left( \int_0^1 f(x)^2 dx \cdot \int_0^1 1^2 dx \right)^2$$

$$\geq \mathbb{E}(\lambda)^2 \left( \left( \int_0^1 f(x)dx \right)^2 \right)^2$$

$$(Cauchy's\ Inequality)$$

$$= \mathbb{E}(\lambda)^2$$

The equality condition of the first inequality is:

$$\int_x^{x+\mathbb{E}(\lambda)} f(u)du \sqrt{f(x)} / \sqrt{f(x)} \equiv C$$

$$\iff \int_x^{x+\mathbb{E}(\lambda)} f(u)du \equiv C$$

$$\iff \mathbb{E}(\lambda) \cdot f(x) \equiv C$$

$$for\ x \in [0,1],\ where\ C\ is\ a\ Constant$$

Considering that generally $\mathbb{E}(\lambda) \neq 0$ and

$$\int_0^1 f(x)dx = 1,$$

we further derive that $f(x) \equiv 1\ for\ x \in [0,1]$.

The equality condition of the second inequality is:

$$f(x)/1 \equiv C$$

$$for\ x \in [0,1],\ where\ C\ is\ a\ Constant$$
$$\implies f(x) \equiv 1\ for\ x \in [0,1]$$

Thus, the overall equality condition of the main derivation is $f(x) \equiv 1\ for\ x \in [0,1]$.

# B    Detailed Experimental Information

## B.1    Detailed Datasets Statistics

The detailed statistics of annotated benchmark datasets we evaluate in the experiments are shown in Table 5 and Table 6. For datasets that contains more than one qualities, we select the quality closest to the overall quality for evaluating. For DSTC10 datasets, we also report the average results on all the 11 qualities as this is the final evaluating indicator of DSTC10 track5 subtask1[5].

Table 5: Quality-annotated datasets used in experiments and the corresponding SOTA methods.

| Dataset | SOTA method |
|---|---|
| FED (Mehri and Eskénazi, 2020a) | DynalEval (Zhang et al., 2021a) |
| GRADE (Huang et al., 2020)
- ConvAI2-Generator (GCG) - DailyDialog-Generator (GDG)
- Empathetic-Generator (GEG) - ConvAI2-Ranker (GCR)
- DailyDialog-Ranker (GDR) - Empathetic-Ranker (GER) | GRADE |
| USR (Mehri and Eskénazi, 2020b)
- TopicalChat (UT) - Persona (UP) | USR |
| DSTC6 (Hori and Hori, 2017) | Deep AM-FM (Zhang et al., 2020a) |
| Predictive Engagement (PE) (Ghazarian et al., 2020) | USL-H (Phy et al., 2020) |
| DSTC10 (Zhang et al., 2021b)
- JSALT - ESL - NCM - TopicalChat (DT) - Persona (DP) | MME-CRS (Zhang et al., 2022b) |

## B.2    Detailed Experimental Setup

We set learning rate as 5e-5, batch size as 32. AdamW Loshchilov and Hutter (2017) is used as the optimizer with $\beta_1 = 0.9$, $\beta_2 = 0.999$ and eps = 1e-6. We use linear-scheduled learning rate strategy with decay = 0.02. We train the model for 40 epochs (except for GRADE and USR) and take the checkpoint of the last epoch. $\beta$ of DP loss finally converges to 100. All experiments are based on a single NVIDIA GeForce RTX 3090 GPU with 24GB memory. We use the following Python code to calculate Spearman correlations and corresponding p-value: *from scipy import stats; spearmanr, p_value = stats.spearmanr(score, label)*. The p-value here roughly indicates the probability of an uncorrelated system producing datasets that have a Spearman correlation at least as extreme as the one computed from these datasets.

We directly used the loss on the Dailydialog validation set as a criterion for hyperparameter selection except for $(s, l)$. As for $(s, l)$, we choose $(3,7)$ in our setting as it can bring the highest $Uniformity$.

## B.3    Detailed Experimental Results

Apart from Spearman correlations, we also test TDEMs on Pearson correlations to gain a more comprehensive understanding of BCR framework. As some research have not reported Pearson correlations results of their methods, comparison can only be performed on 10 datasets. As shown in Table 7, BERT-base applying our BCR framework obtains the highest correlations with human judgements on 8 of 10 datasets. This further demonstrates the excellent effectiveness of BCR framework.

---

[5]https://dstc10.dstc.community/home

Table 6: Detailed statistics of quality-annotated datasets used in experiments. **Num. Samples** column refers to the number of samples of the corresponding dataset. **Num. Qualities** column refers to the number of annotated qualities of the corresponding dataset. **Eva. Quality** column refers to the quality we evaluate in the experiments.

| Dataset | Num. Samples | Num. Qualities | Eva. Quality |
|---------|--------------|----------------|--------------|
| FED | 375 | 9 | Overall |
| GRADE - ConvAI2-Generator | 150 | 1 | Coherence |
| GRADE - ConvAI2-Ranker | 150 | 1 | Coherence |
| GRADE - DailyDialog-Generator | 150 | 1 | Coherence |
| GRADE - DailyDialog-Ranker | 150 | 1 | Coherence |
| GRADE - Empathetic-Generator | 150 | 1 | Coherence |
| GRADE - Empathetic-Ranker | 150 | 1 | Coherence |
| USR - TopicalChat | 300 | 6 | Overall |
| USR - Persona | 240 | 6 | Overall |
| DSTC6 | 40000 | 1 | Overall |
| Predictive Engagement | 600 | 1 | Engagement |
| DSTC10 - JSALT | 741 | 1 | Appropriateness |
| DSTC10 - ESL | 1242 | 1 | Appropriateness |
| DSTC10 - NCM | 2461 | 1 | Appropriateness |
| DSTC10 - TopicalChat | 4500 | 4 | Appropriateness |
| DSTC10 - Persona | 5000 | 4 | Appropriateness |

Table 7: Pearson correlations between TDEMs and human judgements on benchmark datasets. BERT refers to our baseline model. The results of SOTA TDEMs come from the corresponding papers. All results are statistically significant (p-value > 0.05).

| Method | FED | GCG | GCR | GDG | GDR | GEG | GER | PE | DSTC6 |
|--------|-----|-----|-----|-----|-----|-----|-----|-----|-------|
| BERT | 0.205 | 0.616 | 0.491 | 0.260 | 0.261 | 0.167 | 0.323 | 0.612 | 0.234 |
| SOTA | - | 0.606 | **0.535** | 0.368 | 0.261 | 0.257 | 0.375 | **0.688** | **0.326** |
| BERT+BCR | **0.307** | **0.636** | 0.510 | **0.377** | **0.444** | **0.348** | **0.399** | 0.666 | 0.273 |

| Method | UT | UP | JSALT | ESL | NCM | DT | DP | DSTC10 | **Average** |
|--------|-----|-----|-------|-----|-----|-----|-----|--------|---------|
| BERT | 0.339 | 0.405 | 0.171 | 0.403 | 0.246 | 0.297 | 0.412 | 0.281 | 0.336 |
| SOTA | 0.422 | 0.411 | - | - | - | - | - | - | - |
| BERT+BCR | **0.437** | **0.452** | **0.266** | **0.519** | **0.323** | **0.341** | **0.428** | **0.318** | **0.414** |

## B.4 Human Evaluation of Coherence Comparison

We conduct human evaluation to examine the coherence level of samples generated by TSR. Compared data augmentation methods include: (a) **GRADE** (Huang et al., 2020) applies both lexical sampling and embedding-based sampling to obtain negative samples; (b) **PONE** Lan et al. (2020) applies EDA (Wei and Zou, 2019) and generative dialogue model to obtain positive samples. For a given conversation, we use the above two data augmentation methods and medium coherence samples generating strategies (positive and negative) to obtain four samples respectively and ask five human annotators to sort them based on the response coherence. We choose researchers in the field of dialogue as annotators because this work requires certain professional quality. The experiment is conducted on 100 conversations of DailyDialog dataset. We calculate the average ranking of each type of the above methods. The average Fleiss's kappa of annotators is 0.63, indicating annotators have reached relatively strong agreement. The result of coherence ranking is PONE-positive (1.3) >

TSR-positive (1.8) > TSR-negative (3.3) > GRADE-negative (3.6), indicating that TSR can provide stronger training signals with medium coherence compared with general data augmentation methods.

## C More Experimental Results

### C.1 Impact of Training Dataset

We also evaluate the generalization of BCR on different training datasets. We choose DailyDialog (Li et al., 2017), Empathetic (Rashkin et al., 2019), Topical (Gopalakrishnan et al., 2019) as training datasets and test them on DSTC10 dataset respectively. As shown in Table 8, BCR framework brings significant progress to BERT in all settings, which further demonstrates the strong generalization of BCR. We also notice that TDEM trained on DailyDialog performs better than other datasets, which explains why most TDEMs choose this dataset as training set.

### C.2 BCR for Pre-training

In the future, supervised TDEM will probably achieve higher consistency with human judgements. But self-supervised TDEM can still serve as pre-trained model to help supervised TDEM converge faster to better performance under limited labeled data volume conditions. Thus, we compared the performance of supervised TDEM with TDEM that first applied BCR pre-training and then underwent supervised fine-tuning. Specifically, we split FED dataset into train set and test set in a ratio of 7:3 and the train set is served for supervised learning. We finetuned vanilla BERT and BERT-BCR trained on DailyDialog on the labeled train set and test them respectively. Supervised TDEM converged on the $5^{th}$ epochs and attained 0.240 Spearman correlations while TDEM underwent both BCR pre-training and supervised finetuning converged on the $3^{th}$ epochs and attains 0.372 Spearman correlations. This preliminary experiment may inspire scholars to research and apply pre-trained TDEMs to improve the performance of reward models, thus attaining better LLMs.

### C.3 Comparing with GPT4

Recently, some automatic evaluation methods based on LLMs (e.g., GPT-4) (Liu et al.; Wang et al., 2023) have been proposed to explore new paradigms for addressing TDEM. We have compared BCR with Liu et al. in Table 9. The results of Liu et al. is obtained with GPT-4 (version: May 15, 2023). Even in the case of n=20 (self-consistency), BCR+BERT is still able to achieve better average results than G-EVAL. LLM based methods also have the following drawbacks: higher expenses; more time costs; much more computational cost and parameters; bias towards specific content. Therefore, how to effectively combine BCR and LLM is one of our future research directions.

### C.4 Transferability to Other Tasks

We further discuss the possibility of migrating BCR to other self-supervised regression tasks. BCR can be transferred to other tasks, such as Factuality (Falke et al., 2019; Wang et al., 2020), Data to Text (Mairesse et al., 2010), Text Summarization (Fabbri et al., 2021), etc.

We take Text Summarization task for an example. As for TSR, in the first step, we can obtain positive samples with medium coherence by randomly deleting or replacing sentences in the text part of the positive samples; we can also obtain negative samples with medium coherence by randomly deleting or replacing sentences in the summarization part of the positive samples. Based on this, TSR can be implemented according to Eq. (5). As for DP loss, we directly use Eq. (8) to train BCR. It turns out that BERT with DP loss can obtain 0.243 Spearman correlations and -0.013 $Diff$, both better than Triplet loss (0.216 Spearman correlations and -0.030 $Diff$) and MSE loss (0.205 Spearman correlations and -0.039 $Diff$).

## D Further Discussion

In the task of automatic dialogue evaluation, the mainstream approach is to utilize the SSL framework. This is because constructing a manually annotated training dataset for automatic dialogue evaluation is highly challenging. For instance, we once attempted to annotate a evaluation dataset for TDEM,

Table 8: Spearman correlations of BERT and BERT+BCR trained on different datasets.

| Training dataset | Method | JSALT | ESL | NCM | DT | DP | DSTC10 |
|---|---|---|---|---|---|---|---|
| DailyDialog | BERT | 0.166 | 0.345 | 0.253 | 0.220 | 0.401 | 0.255 |
| | BERT+BCR | **0.272** | **0.531** | **0.336** | **0.305** | **0.428** | **0.316** |
| Empathetic | BERT | 0.149 | 0.301 | 0.129 | 0.140 | 0.271 | 0.157 |
| | BERT+BCR | **0.215** | **0.375** | **0.161** | **0.166** | **0.304** | **0.191** |
| Topical | BERT | 0.082 | 0.201 | 0.113 | 0.197 | 0.277 | 0.152 |
| | BERT+BCR | **0.124** | **0.240** | **0.135** | **0.253** | **0.309** | **0.194** |

Table 9: Spearman correlations of BCR and G-EVAL on GRADE datasets. n represents GPT4 generating n scores for each sample and taking the average (self-consistency).

| Method | GDG | GDR | GCG | GCR | GEG | GER | Avg. |
|---|---|---|---|---|---|---|---|
| G-EVAL n=1 | 0.427 | 0.422 | 0.501 | 0.464 | 0.267 | 0.370 | 0.408 |
| G-EVAL n=20 | **0.448** | 0.448 | 0.513 | **0.559** | 0.296 | **0.398** | 0.443 |
| BERT+BCR | 0.416 | **0.487** | **0.642** | 0.519 | **0.341** | 0.372 | **0.463** |

consisting of 600 instances. Each instance comprises a context and multiple responses. Annotators were tasked with assigning scores to these responses individually. Each instance was annotated by 5 individuals to reduce noise. Given the longer dialogue contexts, it took approximately 2 minutes for a single annotator to complete each instance. After data cleaning (removing samples with low internal consistency), approximately 500 instances remained. As a result, annotating these 500 instances required a total of around 100 hours of annotation time. Based on this estimation, if we were to annotate a training set of the size of DailyDialog (13118 samples), we would need approximately 2623 hours of annotation time. Even so, this still does not guarantee that such a training set would exhibit strong generalization across multiple dialogue domains. The requirement for high inter-annotator consistency and the characteristic of longer dialogue contexts significantly elevate the annotation cost of dialogue evaluation training sets. We believe that achieving better evaluation performance requires a self-supervised (BCR) pre-training approach to impart the evaluation model with generalized assessment capabilities. Subsequently, fine-tuning can be carried out with limited annotated training data specific to the data domain. Thus, we look forward to the emergence of high-quality dialogue evaluation training sets in the future.

