# OpenReview forum: "Better Correlation and Robustness: A Distribution-Balanced Self-Supervised Learning Framework for Automatic Dialogue Evaluation"
_NeurIPS.cc/2023/Conference — NeurIPS 2023 poster_

### Official Review · Reviewer_WZUk · 2023-07-04

**Soundness:** 2 fair
**Presentation:** 3 good
**Contribution:** 2 fair
**Rating:** 5
**Confidence:** 3

**Summary:**

This paper proposes a new framework to enhance prior TDEMs by utilizing a self-supervised learning approach. The authors introduce a simple and intuitive framework called BCR. The BCR framework employs two approaches: TSR, which reconstructs the training set using a heuristic to balance the coherence distribution, and DP loss, which ensures uniform score distribution. Experimental results demonstrate that their framework improves the correlation between machine judgment and human judgment, outperforming state-of-the-art TDEM approaches when applied to a vanilla BERT-based model.

**Strengths:**

I believe the motivation of the paper is clearly stated, though I feel confused about the key assumption of this paper (described in Weaknesses). The authors provide sufficient illustrations to demonstrate the limitations of current TDEM approaches. They propose an intuitive approach to address these limitations and support their claims with experimental results. Although their approach may not be technically novel, their insights are meaningful.

**Weaknesses:**

- The main assumption of this paper is that the test set consists of a uniformly distributed coherence, but I am not clear about the justification for this statement. It would be helpful if the authors could provide more background information on this assumption.

- The keyword used in this work is "medium coherence," which is initially challenging to understand. Both "medium" and "coherence" are ambiguous terms. What is the definition of "medium" and "coherence"? This lack of clarity weakens the soundness of the paper. In prior works that address automatic dialogue evaluation, most provide clear definitions of metrics such as coherence, interestingness, etc.

- Some details about the experiments are insufficient and unclear. The paper would be more robust if the authors could clarify these aspects. I will provide detailed questions regarding the experimental details below (Questions).

Though, I am willing to raise the current rating if the authors provide sufficient explanations about these concerns.

**Questions:**

These questions are listed in order of priority:

- How was hyperparameter tuning performed? How were the learning rate, batch size, number of epochs, s, and l for DP loss selected?
  - I could not find details about the hyperparameter tuning, and if the experiments were not conducted based on a train/validation split, then the experimental results may not be reliable.
  - If fair hyperparameter tuning was conducted, what criteria were used to select the hyperparameters? Validation loss is not directly suitable for selecting hyperparameters.
- Figure 7 & Sec 5.2.2: I am having trouble understanding the explanation in Section 5.2.2; aren't the values in Figure 7 only represent "Diffs," not "Spearman correlations"?
- Table 2: In the Spearman correlations of TSR, the table shows that Polarized exhibits better correlations compared to medium. Isn't it natural? The Polarized samples indicate that the coherence is much easier to distinguish. More detailed explanations about the experiments would help in understanding the results.
- Table 3: The explanation about Example 2 is unclear; "BCR can bring better correlations with human judgments on medium coherence samples." Can you provide further explanation about this? When comparing USR and USR+BCR, I only observe a small gap between them.



**Limitations:**

I think the authors have addressed their limitations.

---

> ### Author Rebuttal · Authors · 2023-08-07
>
> We thank the reviewer for the positive comments as well as constructive suggestions! Below, we discuss each of the reviewer's concerns in detail.
> > **The main assumption of this paper is that the test set consists of a uniformly distributed coherence, but I am not clear about the justification for this statement. It would be helpful if the authors could provide more background information on this assumption.**
>
> In the field of dialogue evaluation, the datasets proposed in previous works are typically balanced (e.g., "the distributions of human judgements are balanced from score 1 to 5" from [1]). Intuitively, evaluation sets with more balanced score distributions are better suited for comprehensively assessing the performance of different dialogue models. (Imagine a test set with only 1-point and 5-point ratings, as opposed to evenly distributed ratings of 1, 2, 3, 4, and 5.)
>
> > **The keyword used in this work is "medium coherence," which is initially challenging to understand. Both "medium" and "coherence" are ambiguous terms. What is the definition of "medium" and "coherence"? This lack of clarity weakens the soundness of the paper. In prior works that address automatic dialogue evaluation, most provide clear definitions of metrics such as coherence, interestingness, etc.**
>
> Here, "medium coherence" refers to samples with human ranking scores falling within the range [0.25, 0.75], indicating a level of consistency between the responses and the preceding context that lies between strong and weak. We appreciate your reminder, and we will incorporate the corresponding definition in the upcoming revisions.  Thank you for your valuable suggestion.
>
>
> > **How was hyperparameter tuning performed? How were the learning rate, batch size, number of epochs, s, and l for DP loss selected?**
>
> We use the validation loss  and $Uniformity$ (only for ($s$,$l$)) as criterions for hyperparameter selection. As you mentioned, validation loss is not directly suitable for selecting hyperparameters. However, In order to align with the testing performance of previous research work, we did not split the evaluation set we used into a validation / test set (which would result in incomplete testing data). Instead, we directly used the loss on the Dailydialog validation set as a criterion for hyperparameter selection except for ($s$, $l$). As for ($s$, $l$), we choose (3,7) in our setting as it can bring the highest $Uniformity$ (line 273). As a matter of fact, if there is a validation set with high quality human annotations available (outside of evaluation set we used), the performance of model would be further improved after hyperparameter tuning. See Figure 8, if the values of (s, l) are chosen as (3,8), better results in both consistency and robustness can be achieved compared to the (3,7) we selected according to $Uniformity$. However, in order to ensure the rigor of the experiment, we did not choose (3,8) as the default option for our (s, l), but instead retained (3,7).
>
> > **Figure 7 & Sec 5.2.2: I am having trouble understanding the explanation in Section 5.2.2; aren't the values in Figure 7 only represent "Diffs," not "Spearman correlations"?**
>
> As illustrated in Figure 7, DSTC10, NCM, DT, DP, ESL, JSALT represent the model's "Spearman Correlations" on the respective datasets. The values under "Diff" (highlighted in red) indicate the model's average performance of Diff metric on all these datasets. We apologize for any confusion caused by the unclear description and we will clarify this point in the caption of Figure 7 in our revised version. Thanks for your reminder.
>
> > **Table 3: The explanation about Example 2 is unclear; "BCR can bring better correlations with human judgments on medium coherence samples." Can you provide further explanation about this? When comparing USR and USR+BCR, I only observe a small gap between them.**
>
> Regarding Example 2 in Table 3 (U1: Why aren't you eating anything else? U2: Well, fruits and vegetables are very healthy. R: What kind of vegetables do you want to eat?), the response "R: What kind of vegetables do you want to eat?" is considered as medium coherence by human evaluators (with a corresponding ranking of 0.45). In comparison to USR (which has a ranking of 0.87), the alignment between USR+BCR and human evaluators is higher in consistency (with a ranking of 0.47, approaching the human ranking of 0.45), which suggests the conclusion ("BCR can bring better correlations with human judgments on medium coherence samples.").
>
> > **Table 2: In the Spearman correlations of TSR, the table shows that Polarized exhibits better correlations compared to medium. Isn't it natural? The Polarized samples indicate that the coherence is much easier to distinguish. More detailed explanations about the experiments would help in understanding the results.**
>
> We calculated the Spearman scores for the "polarized" and "medium" samples separately as follows: For the polarized samples, we computed the model's Spearman performance on subsets of samples with labels within [0, 0.25] and [0.75, 1] respectively, and then averaged these values to obtain the polarized score. For the medium sample, we considered the model's Spearman performance on samples within the [0.25, 0.75] score range, and referred it as the medium score. In other words, we evaluated the model's performance for polarized and medium samples within their corresponding score ranges. Therefore, "Polarized exhibits better correlations compared to medium" is not a natural conclusion.
>
> If we have any unclear explanations, please feel free to ask us any questions. We look forward to helping you solve your confusion!
>
> [1] GRADE: Automatic Graph-Enhanced Coherence Metric for Evaluating Open-Domain Dialogue Systems

---

> > ### Comment · Reviewer_WZUk · 2023-08-13
> >
> > I appreciate for your explanations about the reviews. and I update my rating from 4 to 5.

---

> > > ### Author Response · Authors · 2023-08-13
> > > **Thanks for Raising Rating**
> > >
> > > We sincerely thank you for your recognition of BCR!

---

### Official Review · Reviewer_yNip · 2023-07-09

**Soundness:** 3 good
**Presentation:** 3 good
**Contribution:** 3 good
**Rating:** 6
**Confidence:** 4

**Summary:**

This paper introduces a turn-level dialogue evaluation model (TDEM) and focuses on the self-supervised learning (SSL) framework, which has recently demonstrated state-of-the-art (SOTA) performance in open-domain dialogue evaluation.

The authors identify two potential problems with existing SSL methods for training TDEMs. Firstly, these methods tend to have low correlations with humans on medium coherence samples. Secondly, the SSL approach leads to a nonuniform score distribution in TDEM.
In response to these drawbacks, the paper presents the "Better Correlation and Robustness (BCR)" method, which enables distribution-balanced self-supervised learning for TDEM.
This method includes two techniques, TSR and DP, and is applied during the SSL training of TDEM. TSR is responsible for reconstructing a coherence distribution balanced training set with a continuous label domain. DP loss, on the other hand, adjusts adaptively based on the score distribution estimated by kernel density estimation to achieve a more uniform distribution.

The authors conducted experiments on 17 benchmark datasets and found that the vanilla BERT-base with the proposed BCR method outperforms current SOTA methods by an average of 11.3%.
Additionally, the experiments also demonstrate the generalization ability of the proposed method by combining the proposed method with the current SOTA methods.

**Strengths:**

* The motivation behind this paper is well-founded as it addresses a relevant and significant research gap.

* The paper is generally easy to comprehend (although the mathematical components may require additional effort to fully grasp.)

* The proposed method is captivating and demonstrates novelty, offering a fresh perspective on TDEMs.

* The experimental results presented in the paper seem robust and compelling, providing substantial evidence to support the efficacy of the proposed approach (if we believe that the results are correct).

**Weaknesses:**

* This paper explores exclusively the narrow topic of turn-level dialogue evaluation. I understand that dialogue evaluation is a crucial research topic in the AI research field. While the paper demonstrates improved performance of the BCR method compared to current state-of-the-art (SOTA) methods in TDEMs, it lacks a discussion on the transferability and practicality of the proposed approach to other similar tasks in the literature.
While this may not be a critical flaw that warrants an immediate rejection of the paper from the conference, it may pose a presentational drawback compared with other papers that explicitly discuss the potential broader applicability across various AI research areas.

* Another limitation is the lack of discussion on the computational costs associated with implementing the proposed method.

* The impact of the proposed method is unclear to me at least in the current status: first, as the Spearman correlation scores for all the methods, including the BCR method, seem to fall within a similar range (around 0.4), which is generally considered to indicate a low or moderate correlation.
Second, regarding the statistical test, The captions in Tables 1 and 4 say "All results are statistically significant (p-value > 0.05)." However, there is no explicit procedure how the authors confirmed this statistical significance. It should be explained in the paper, at least in the appendix.

**Questions:**

* I am not very knowledgeable about dialogue evaluation, but one potential solution to address the scarce issue of score distribution could be to increase the amount of human evaluation data for the range of scores that are currently scarce, such as medium correlation scores that are discussed in this paper. However, the paper does not delve into the reasons why this straightforward approach cannot be implemented. Nowadays, acquiring a sufficient amount of human judgement data is not particularly challenging, so it may be worthwhile to explore this avenue further. Additionally, it is intriguing to consider the improvements achieved through the proposed method in terms of their equivalence to the number of human annotated samples. If this number is substantial and difficult to obtain, it could further enhance the effectiveness of the proposed method.

* As explained in the weaknesssection

**Limitations:**

* Given that this paper focuses on evaluating dialogue systems, there is a potential risk of evaluating systems in an inaccurate manner. This risk is evident from the relatively not enough high correlation observed between the model's outputs and human judgments.

---

> ### Author Rebuttal · Authors · 2023-08-09
>
> We thank the reviewer for the positive comments as well as constructive suggestions! Below, we discuss each of the reviewer's concerns in detail.
>
> > **This paper explores exclusively the narrow topic of turn-level dialogue evaluation. I understand that dialogue evaluation is a crucial research topic in the AI research field. While the paper demonstrates improved performance of the BCR method compared to current state-of-the-art (SOTA) methods in TDEMs, it lacks a discussion on the transferability and practicality of the proposed approach to other similar tasks in the literature.**
>
> BCR can be widely applied to various other tasks, such as evaluation of summarization, QA and so on. To demonstrate that, we choose summarization evaluation set SummEval [1] (relevance) and compared the performance of BERT with MSE,  Triplet, and DP loss respectively. As shown below:
> |Loss|Spearman|$Diff$|
> |:--|:--:|:--:|
> |MSE|0.205| -0.039|
> |Triplet|0.216|-0.030|
> |DP|**0.243**|**-0.013**|
>
> As can be observed, compared to the commonly used losses for regression tasks, DP loss is capable of significantly enhancing the model's consistency (higher Spearman) and robustness (higher $Diff$) . We will further validate its effectiveness on additional tasks in the future. Thank you for your suggestions.
>
> > **Another limitation is the lack of discussion on the computational costs associated with implementing the proposed method.**
>
> Below is a comparison of BCR with respect to parameter count, training costs, and computational costs to the methods we have examined.
>
> |Method|BCR+BERT|USR|GARDE|MME-CRS|
> |:-|:-:|:-:|:-:|:-:|
> |Parameter Counts (M)|110| 373.8|469|>435.2|
> |Training Costs (hours)|~2|~6|~7|>4|
> |Computational Costs|$T_{base}$|3*$T_{base}$|$T_{base}$+$T_{GNN}$|5*$T_{base}$|
>
> $T_{base}$ denotes the computational cost for a single pre-trained fine-tuned model in BCR and $T_{GNN}$ denotes graph neural network. We will incorporate this section of discussion in the revised version, thanks.
>
> > **The impact of the proposed method is unclear to me at least in the current status: first, as the Spearman correlation scores for all the methods, including the BCR method, seem to fall within a similar range (around 0.4), which is generally considered to indicate a low or moderate correlation.**
>
> Prior to BCR, SOTA automatic evaluation methods exhibited lower Spearman Correlations (seen Table 1), due to the challenging nature of dialogue evaluation, even for humans (0.63 inter-annotator correlation in [2]). The significance of our work lies in uncovering the inherent issue of uneven training data distribution in TDEM and theoretically demonstrating the relationship between the scoring distribution of evaluation models and robustness. By employing TSR and DP-loss, we elevated the BERT model to the latest SOTA performance and even surpassed GPT4 (see response to Reviewer z9qv), thereby fostering the advancement of future dialogue evaluation tasks.
>
> > **Second, regarding the statistical test, The captions in Tables 1 and 4 say "All results are statistically significant (p-value > 0.05)." However, there is no explicit procedure how the authors confirmed this statistical significance. It should be explained in the paper, at least in the appendix.**
>
> We apologize that we made a typo here that the p-value should be less than 0.05 (generally less than 1e-3 in fact), and we will make corrections in the revised version. According to the definition of the p-value [3], when the p-value is less than the pre-set significance level (usually 0.05 in most papers), it implies that we have sufficient evidence to consider the observed data statistically significant.
>
> > **One potential solution to address the scarce issue of score distribution could be to increase the amount of human evaluation data for the range of scores that are currently scarce, such as medium correlation scores that are discussed in this paper. However, the paper does not delve into the reasons why this straightforward approach cannot be implemented. Nowadays, acquiring a sufficient amount of human judgement data is not particularly challenging, so it may be worthwhile to explore this avenue further. Additionally, it is intriguing to consider the improvements achieved through the proposed method in terms of their equivalence to the number of human annotated samples. If this number is substantial and difficult to obtain, it could further enhance the effectiveness of the proposed method.**
>
> We highly acknowledge the potential of mitigating the issue of imbalanced training data by augmenting human-annotated data. But this has consistently been challenging. On one hand, extensive annotation incurs high costs in terms of both labor and time. On the other hand, achieving good consistency among human annotators is crucial and hard. This is also why a public labeled training set is practically non-existent in the dialogue evaluation field at present, except for the closed source ADEM[2] dataset in 2017.
>
> We believe that achieving better evaluation performance requires a self-supervised (BCR) pre-training approach to impart the evaluation model with generalized assessment capabilities. Subsequently, fine-tuning can be carried out with limited annotated training data specific to the data domain.  In Appendix C.2, we measured the effects of fine-tuning BERT+BCR compared to directly fine-tuning BERT. This confirms that the training paradigm of SSL pre-training followed by supervised fine-tuning yields better results than direct supervised training (Spearman 0.372 > 0.240). Hence, exploring automatic evaluation under the SSL paradigm is of paramount significance. We greatly appreciate your interesting suggestion, and we look forward to completing this verification when a publicly available training set appears.
>
> [1] SummEval: Re-evaluating Summarization Evaluation
>
> [2] Towards an Automatic Turing Test: Learning to Evaluate Dialogue Responses
>
> [3] Statistical Methods for Research Workers

---

> > ### Comment · Reviewer_yNip · 2023-08-20
> >
> > I apologize for the late in, and I appreciate the authors taking the time to address my questions and concerns.
> > Below are my follow-up comments in response to the authors' feedback.
> >
> > #### *** 1, Transferability to other similar tasks
> > I appreciate the authors including the results of the summarization tasks.
> > I found the dynamic penalty (DP) method can work on similar evaluation tasks.
> > A discussion regarding the method's transferability to comparable tasks, for instance, in a potential section 5.5, would greatly enhance the utility of the proposed approach.
> > One other question is, if my understanding is correct, the proposed method consists of two parts, TSR and DP.
> > it would enhance the paper if the authors could provide examples of other evaluation tasks or scenarios where the TSR-only or TSR and DP combination also contributes.
> >
> >
> > #### *** 2, Computational cost
> > I was concerned about whether the proposed method might necessitate a higher computational expense for improved results.
> > The inclusion of experimental results from a computational cost standpoint assuages this concern.
> > I suggest incorporating the computational cost discussion into the paper,  at least within Appendix.
> >
> >
> > #### *** 3, The meaning of Spearman correlation scores
> > Thank you for the response.
> > I understand that the proposed method improved the performance of Spearman’s correlation scores with human judgment.
> > Nevertheless, as the absolute value of 0.4 for Spearman’s correlation score is not typically considered high, I questioned whether we could trust the scores provided by the proposed method in real dialogue evaluation scenarios.
> > However, I concede that imposing such a requirement might be excessive.
> > I agree with the authors' claim that the results of the proposed method have an impact on the community.
> >
> >
> > #### *** 4,  Statistical test
> > Thank you for the correction of the erroneous statement.
> > My subsequent question is what types of statistical test (e.g., the t-test, Spearman's r, and Wilcoxon's signed rank test) is conducted to compute the p-value and the procedure for calculating it for "All results are statistically significant."
> > I am concerned about whether the p-values were appropriately computed to avoid a typical misuse of statistical tests since no clear procedure was revealed.
> >
> >
> > #### *** 5,  Comparing the impact with human annotation data
> > Thank you for the comment.
> > I am still curious whether the human annotation cost is realy impractical or not, as the authors stated. Maybe I am wrong, but as the authors agreed, no one has ever deeply investigated this point.
> > That is why I said it would be better to confirm (if the authors could) to support the background of this paper and the proposed method.
> > Anyway, this is somewhat not a “must-have” but a “better-to-have” point; thus, this never be a reason to reject the paper.

---

> > > ### Author Response · Authors · 2023-08-20
> > >
> > > We sincerely appreciate your thorough review and constructive feedback. Below is our response:
> > >
> > > > **Transferability to other similar tasks**
> > >
> > > The main idea of TSR is to reconstruct the dataset with an uneven distribution in the self-supervised regression task, enabling the model to obtain balanced supervision signals. TSR mainly includes two steps: The first step is to construct data in the data domain which is lacking in specific tasks (such as data with medium consistency in this paper), and the second step is to denoise the labels of this part of data using the model's prediction (as equation 5 in this paper).
> > > As a matter of fact, TSR can be transferred to other tasks, such as Factuality [1][2], Data to Text [3], Text Summarization [4], etc. Taking Text Summarization as an example, in the first step, we can obtain positive samples with medium consistency by randomly deleting or replacing sentences in the text part of the positive samples; we can also obtain negative samples with medium consistency by randomly deleting or replacing sentences in the summarization part of the positive samples. Based on this, TSR can be implemented according to equation 5.
> > >
> > > > **Computational cost**
> > >
> > > Thank you again for your valuable suggestion, and we will incorporate it into the revised version.
> > >
> > > > **The meaning of Spearman correlation scores**
> > >
> > > In the final dialogue model evaluation stage, we recommend using humans for evaluation to obtain more effective results. While during the development phase, dialogue evaluation models (such as BCR) with higher consistency with human judgments can be used to replace traditional metrics like BLEU for better hyperparameter tuning [5].
> > >
> > > > **Statistical test**
> > >
> > > Spearman's r is used to calculate p-value following mainstream works [6][7] in TDEM. Specifically, we use the following Python code for the calculation of spearmanr and p-value:
> > >
> > > **from scipy import stats**
> > >
> > > **spearmanr, p_value = stats.spearmanr(score, label)**
> > >
> > > The p-value here roughly indicates the probability of an uncorrelated system producing datasets that have a Spearman correlation at least as extreme as the one computed from these datasets.
> > >
> > > > **Comparing the impact with human annotation data**
> > >
> > > As we mentioned earlier, the fact that most mainstream TDEMs adopt the SSL framework and only ADEM dataset from 2017 is the only labeled dataset can reflect the difficulty in obtaining human annotation datasets to some extent.
> > >
> > > Another study we conducted concurrently may provide a detailed illustration of this matter. In this study, we introduced an evaluation dataset for TDEM, consisting of 600 instances. Each instance comprises a context and multiple responses. Annotators were tasked with assigning scores to these responses individually. Each instance was annotated by 5 individuals to reduce noise. Given the longer dialogue contexts, it took approximately 2 minutes for a single annotator to complete each instance. After data cleaning (removing samples with low internal consistency), approximately 500 instances remained. As a result, annotating these 500 instances required a total of around 100 hours of annotation time. Based on this estimation, if we were to annotate a training set of the size of DailyDialog (13118 samples), we would need approximately 2623 hours of annotation time. Even so, this still does not guarantee that such a training set would exhibit strong generalization across multiple dialogue domains.
> > >
> > > The requirement for high inter-annotator consistency and the characteristic of longer dialogue contexts significantly elevate the annotation cost of dialogue evaluation training sets. We look forward to the emergence of high-quality dialogue evaluation training sets. We hope that this fact may further address your concern.
> > >
> > > [1] Ranking Generated Summaries by Correctness: An Interesting but Challenging Application for Natural Language Inference
> > >
> > > [2] Asking and Answering Questions to Evaluate the Factual Consistency of Summaries
> > >
> > > [3] Phrase-Based Statistical Language Generation Using Graphical Models and Active Learning
> > >
> > > [4] SummEval: Re-evaluating Summarization Evaluation
> > >
> > > [5] A Comprehensive Assessment of Dialog Evaluation Metrics
> > >
> > > [6] GRADE: Automatic Graph-Enhanced Coherence Metric for Evaluating Open-Domain Dialogue Systems
> > >
> > > [7] USR: An Unsupervised and Reference Free Evaluation Metric for Dialog Generation

---

> > > > ### Comment · Reviewer_yNip · 2023-08-21
> > > >
> > > > I appreciate the prompt response and the tremendous effort to answer my questions and concerns.
> > > > The authors' response to my concerns has been mostly satisfactory.
> > > >
> > > > As the final comment, I encourage the authors to incorporate answers to my concerns discussed here, such as transferability, computational cost, the detailed procedure of the statistical test used in the experiments (this is mandatory, I think), and examples of human annotation cost, into the paper (or, at a minimum, into the Appendix).
> > > > The inclusion of these responses could serve to reinforce the impact of the methods and motivations put forth in this study.
> > > >
> > > > If these responses had been part of the initial submission, I am certain that my review score would have been more favorable.
> > > > If these responses are included in the final version, I would gladly revise my recommendation score upward.
> > > > However, I am unsure whether such an amendment is within the rules of this review process  (meaning the reviewers should score within the contents of the submission version or the reviewers can consider the future revision).
> > > > Thus, I will confirm the reviewer guidelines first (or ask the AC), and if it is permissible, I will pursue this course of action.

---

> > > > > ### Author Response · Authors · 2023-08-21
> > > > >
> > > > > We sincerely appreciate the diligent completion of the review by Reviewer yNip and the valuable suggestions provided that have made our work more comprehensive and persuasive. We have incorporated the content related to these suggestions into the revised version and have sent it to the Chairs through an anonymous link. (According to the meeting requirements, we cannot place this link here)
> > > > >
> > > > > With sincere regards,
> > > > >
> > > > > Authors

---

> ### Author Response · Authors · 2023-08-18
>
> Dear Reviewer yNip,
>
> We would like to thank you again for the time you dedicated to reviewing our paper. We believe that we have addressed your concerns. Since the end of discussion period is getting close and we have not heard back from you yet, we would appreciate it if you kindly let us know of any other concerns you may have, and if we can be of any further assistance in clarifying any other issues.
>
> Thanks a lot again, and with sincerest best wishes
>
> Authors

---

### Official Review · Reviewer_z9qv · 2023-07-14

**Soundness:** 3 good
**Presentation:** 3 good
**Contribution:** 2 fair
**Rating:** 4
**Confidence:** 4

**Summary:**

This paper proposes a distribution-balanced self-supervised learning framework called Better Correlation and Robustness (BCR) to tackle the challenges of weak performance in medium coherence samples and nonuniform score distribution. Specifically, BCR contains an effective training set reconstructing method to provide coherence-balanced training signals and a loss function which can adjust adaptively based the uniformity of score distribution. Experimental results show the superior performance and robustness of the proposed method.

**Strengths:**

1. The proposed method is overall sound and effective. It may be extended to self-supervised learning (SSL) frameworks for other NLP tasks in addition to dialogue evaluation.

2. Extensive experiments on multiple benchmark datasets demonstrate the effectiveness of the proposed method.

**Weaknesses:**

1. The research challenges tackled by this paper are limited to a specific framework (i.e., SSL) for a specific task (i.e., dialogue evaluation). Thus, the contribution is somewhat narrow especially for a general machine learning conference like NeurIPS. The selected SSL models including GRADE and USR in the experiment are both proposed in 2020. As we are already in 2023, I think that SSL may not be the most mainstream method for dialogue evaluation tasks. Unsupervised evaluation methods based on large language models (LLMs) like ChatGPT / GPT-4 have shown impressive performance in the evaluation of various natural language generation models [1]. It’s obvious that the proposed method which needs to train the evaluation model (such as BERT-base) on specific datasets cannot adapt to unsupervised methods based on LLMs. Thus, the technical contribution may be further weakened.

2. Many parts in the proposed method need more intuitive explanations or theoretical support. For example, from Line 197 to Line 200, the authors set many numbers such as the boundary of polarized / centralized interval and the multiple to compute new $\beta$. I wonder whether these numbers are optimal for the method or they are just an empirical try. Also, there are many other important hyper-parameters such as $s$ and $l$ (Equation 8). The authors should further explain how to tune these hyper-parameters.

3. From Table 5, the SOTA baselines in the main results are mostly proposed before 2021, which are outdated. The authors should select recent SOTA baselines for comparison to judge the performance of the proposed method.


[1] GPTEVAL: NLG Evaluation using GPT-4 with Better Human Alignment.


**Questions:**

I have included my questions in the weaknesses part.

**Limitations:**

The authors have adequately addressed the limitations.

---

> ### Author Rebuttal · Authors · 2023-08-08
>
> We thank the reviewer for the valuable comments as well as constructive suggestions! Below, we discuss each of the reviewer's concerns in detail.
>
> > **The research challenges tackled by this paper are limited to a specific framework (i.e., SSL) for a specific task (i.e., dialogue evaluation). Thus, the contribution is somewhat narrow especially for a general machine learning conference like NeurIPS.**
>
> We rigorously proved Theorem 1 through formula derivation and conducted experimental validation. Theorem 1 is universal and makes a significant contribution to enhancing the robustness of regression task prediction models. From this perspective, our contribution is far-reaching. Additionally, efficient and accurate evaluation of LLMs can effectively promote scholars' assessment and research on the performance of LLMs. Therefore, research on automated dialogue evaluation holds significant significance.
>
> > **The selected SSL models including GRADE and USR in the experiment are both proposed in 2020. As we are already in 2023, I think that SSL may not be the most mainstream method for dialogue evaluation tasks. Unsupervised evaluation methods based on large language models (LLMs) like ChatGPT / GPT-4 have shown impressive performance in the evaluation of various natural language generation models [1]. It’s obvious that the proposed method which needs to train the evaluation model (such as BERT-base) on specific datasets cannot adapt to unsupervised methods based on LLMs. Thus, the technical contribution may be further weakened.**
>
> In fact, we are also tracking the unsupervised evaluation methods based on LLM that you mentioned [1] [2]. In order to align with our model (assessing coherence between responses and context), we reproduced the [1] method on the [3] dataset and obtained the following experimental results with GPT-4 (version: May 15, 2023) (We have stored the data generated during this experiment  in an anonymous link and sent it to AC. According to the meeting requirements, we cannot place this link here, but we are willing to make it public in the future):
>
> |Methods|GDG|GDR|GCG|GCR|GEG|GER|Average|
> | :- | :-:| :-: | :-: | :-: | :-: | :-: | :-: |
> |G-EVAL n=1|0.427|0.422|0.501|0.464|0.267|0.370|0.408|
> |G-EVAL n=20|0.448|0.448|0.513|0.559|0.296|0.398|0.443|
> |BCR|0.416|0.487|0.642|0.519|0.341|0.372|**0.463**|
>
> The parameter n in represents GPT4 generating n scores for each sample and taking the average (a type of Self consistency). The experimental results indicate that across 6 test sets, BCR have surpassed G-EVAL by an average of 4.5% (0.02) , achieving a higher level of coherence with humans. In the field of automated evaluation, LLM still cannot fully replace medium-scale fine-tuned models (BCR).
>
> Furthermore, LLM encounters the following issues when performing dialogue automated evaluation tasks:
> - Higher expenses
> - More time costs
> - Much more computational cost and parameters
> - Bias towards specific content (e.g., LLM-generated content tends to receive higher scores compared to human responses [1]).
>
> Therefore,  current LLM based evaluation framework still needs further development. In fact, we believe that a more reasonable approach in the future is to transfer the knowledge of LLM to medium-scale models in an unbiased manner. Under this notion, BCR can serve as an appropriate pre-training or fine-tuning strategy. Hence, BCR holds significant importance both now and in the future.
>
> > **Many parts in the proposed method need more intuitive explanations or theoretical support. For example, from Line 197 to Line 200, the authors set many numbers such as the boundary of polarized / centralized interval and the multiple to compute new $\beta$. I wonder whether these numbers are optimal for the method or they are just an empirical try. Also, there are many other important hyper-parameters such as $s$ and $l$ (Equation 8). The authors should further explain how to tune these hyper-parameters.**
>
>  The boundary of the polarized interval ([0, 0.25] and [0.75, 1]) and the centralized interval ([0.25, 0.75]) is a natural (and only) selection to ensure that both the polarized interval and the centralized interval each occupy half and do not lean towards any extreme. Setting the update threshold to 0.6 is a natural trial value. We chose 0.6 based on two principles: firstly, we aim for a uniformly distributed final score distribution. Therefore, when a certain interval deviates from 0.5, we should update the loss formulation to correct the bias. Hence, this threshold should be close to 0.5. Secondly, we want to avoid updating the loss formulation too frequently, so this threshold should not be too close to 0.5. Therefore, we chose 0.6, believing that further improvement in BCR's effectiveness can be achieved through hyperparameter searching. As for ($s$ and $l$), we choose (3,7) in our setting as it can bring the highest $Uniformity$ (line 273). We demonstrate in Figure 8 that different values of $s$ and $l$ generally achieve good results.
>
> We will incorporate this section of content in the revised version. Thank you for your valuable suggestion.
>
> > **From Table 5, the SOTA baselines in the main results are mostly proposed before 2021, which are outdated. The authors should select recent SOTA baselines for comparison to judge the performance of the proposed method.**
>
> Table 1 includes the first-place solution for the DSTC10 Dialogue Evaluation Track in 2022 (DSTC being a top international competition in the field of dialogue). Furthermore, as mentioned earlier, we have also compared our results with the G-EVAL method (May 2023), which you kindly suggested, and achieved better performance. Therefore, we believe that we have compared BCR to state-of-the-art methods.
>
> [1] GPTEVAL: NLG Evaluation using GPT-4 with Better Human Alignment.
>
> [2] Is ChatGPT a Good NLG Evaluator? A Preliminary Study.
>
> [3] GRADE: Automatic Graph-Enhanced Coherence Metric for Evaluating Open-Domain Dialogue Systems.

---

> > ### Comment · Reviewer_z9qv · 2023-08-21
> > **Response to Rebuttal**
> >
> > Thanks for your rebuttal.
> >
> > As for the first response, I'm not convinced that Theorem 1 is universal enough because it is built based on the formulation in Section 3.1 (i.e., the formulation of SSL). From my understanding, it is hard to transfer this theorem / method to supervised (such as BLEURT [1]) and unsupervised (such as G-EVAL) evaluation metrics, which are more widely applicable.
> >
> > As for the second and fourth response, I think that comparison with up-to-date baselines is important and necessary. Thus, I expect that this part of the rebuttal should be surely included in this paper and may also need another round of peer reviews.
> >
> > [1] BLEURT: learning robust metrics for text generation. ACL 2020.

---

> > > ### Author Response · Authors · 2023-08-21
> > >
> > > Dear Reviewer z9qv,
> > >
> > > We would like to clarify that the proof of Theorem 1 does not rely on the SSL framework (as presented in Section 3.4, we have not included any assumptions based on the SSL framework). Therefore, its applicability encompasses both the unsupervised and supervised paradigms you mentioned. We apologize for not emphasizing this point in the main text, which may have caused confusion for you. As for the methods mentioned in this work (TSR and DP loss), they are designed for the SSL architecture (as indicated by our title), and thus cannot be directly transferred to the unsupervised and supervised paradigms.
> > >
> > > Regarding your suggestion for comparing with up-to-date baselines, we have incorporated this recommendation in the revised version and have included it in an anonymous link sent to the chairs.
> > >
> > > Once again, we sincerely appreciate your valuable suggestions and thorough review.
> > >
> > > Authors

---

> ### Author Response · Authors · 2023-08-18
>
> Dear Reviewer z9qv,
>
> We would like to thank you again for the time you dedicated to reviewing our paper. We believe that we have addressed your concerns. Since the end of discussion period is getting close and we have not heard back from you yet, we would appreciate it if you kindly let us know of any other concerns you may have, and if we can be of any further assistance in clarifying any other issues.
>
> Thanks a lot again, and with sincerest best wishes
>
> Authors

---

### Official Review · Reviewer_7oUE · 2023-07-21

**Soundness:** 3 good
**Presentation:** 3 good
**Contribution:** 3 good
**Rating:** 7
**Confidence:** 4

**Summary:**

This manuscript aims at two problems in turn-level dialogue evaluation models (TDEMs), including lower correlations between human judgments and the self-supervised learning (SSL) framework on medium coherence samples, and the nonuniform score distribution. The authors also provide a theoretical analysis to demonstrate that nonuniform score distribution would hurt the robustness of TDEM. To tackle the problems, this work first presents a training set reconstructing (TSR) method, i.e., constructing hard pos/neg samples, to balance the polarized data distribution, then presents a dynamic penalty loss function (DP loss) to balance the score distribution. The proposed methods are evaluated on 17 benchmark datasets with BERT-based models, and exhibit good performance. More in-depth analyses are also included to verify the effectiveness and generalization of the methods.

**Strengths:**

1. Good motivation and clear problem formulation. The work studies an important and interesting problem in the field of dialogue evaluation.
2. The manuscript is well-organized and clearly presented. It is easy for the readers to follow up.
3. The proposed framework is solid and effective.
3. Comprehensive experiments and evaluations.
The overall quality of this manuscript surpasses the borderline of acceptance.

**Weaknesses:**

1. The novelty is not very significant. The techniques of TSR are commonly used in dialogue systems.
2. I'd like to see the experiments with other PrLMs, apart from BERT.
3. There are seemingly many parameters needing to search, including the initial $\beta$, N in TSR, the values of $s$ and $l$, etc. How do I select the best parameters on different datasets?

**Questions:**

A small question about the TSR part. You exchange the order of U1 and U2 to create a pseudo-positive sample. Do the additional embeddings, like $E_A E_B E_A$, reorder to $E_B E_A E_A$ accordingly or keep it unchanged?

**Limitations:**

The limitations are discussed in Section 6. Authors also provide some reasonable explanations. I did not fund any potential negative societal impact.

---

> ### Author Rebuttal · Authors · 2023-08-07
>
> We thank the reviewer for the positive comments as well as constructive suggestions! Below, we discuss each of the reviewer's concerns in detail.
> > **The novelty is not very significant. The techniques of TSR are commonly used in dialogue systems.**
>
> We highly value your feedback and would also like to provide the following addition: TSR has been demonstrated in the manuscript as an effective method for reconstructing a training dataset with a balanced coherence distribution. Despite its simplicity and intuitiveness, this method, compared to other methods contrasted in the article, exhibits a better ability to address the issue of insufficient moderately consistent samples mentioned in the article. Additionally, we consider the theoretical proof and empirical validation (via DP loss) of Theorem 1 to be innovative and significant (thanks once again for your acknowledgment). We believe that Theorem 1 and the proposed DP loss hold the potential for future applications in other tasks.
>
> > **I'd like to see the experiments with other PrLMs, apart from BERT.**
>
> USR adopts RoBERTa as the base model. We conducted comparative tests on the effectiveness of USR combined with the BCR framework. As shown in Figure 7, USR+BCR outperforms USR in both Spearman correlations and robustness, consistent with the conclusion drawn when using BERT as the base model.
>
> > **There are seemingly many parameters needing to search, including the initial $\beta$, N in TSR, the values of $s$ and $l$, etc. How do I select the best parameters on different datasets?**
>
> We applied the BCR framework on different training sets and presented the results in Appendix C.1. Throughout this process, we maintained the following hyperparameter settings: initial $\beta$ set to 1, N set to 20, and $s$ and $l$ set to 3 and 7, respectively. This set of configurations yielded favorable experimental outcomes across all trials. For a new setting, as $\beta$ can be adaptively adjusted during training, it is not highly sensitive to the initial value; we suggest maintaining N within the range of [10, 30]. For clean dialogue datasets (e.g., Dailydialog), due to the relatively low volume of data with medium coherence, N can be appropriately increased for supplementation. Conversely, for datasets with higher noise levels (e.g., Twitter), where a larger amount of medium coherence data may exists, a slight reduction in N can be considered. As for ($s$ and $l$), we choose (3,7) in our setting as it can bring the highest $Uniformity$ (line 273). If you want to achieve better results for a new setting, we suggest adjusting hyperparameters based on the loss on the validation set and the $Uniformity$ of the model score distribution.
>
> We will incorporate this section of content in the revised version. Thank you for your valuable suggestion.
>
> > **A small question about the TSR part. You exchange the order of U1 and U2 to create a pseudo-positive sample. Do the additional embeddings, like $E_AE_BE_A$, reorder to $E_BE_AE_A$ accordingly or keep it unchanged?**
>
> We keep the additional speaker embeddings unchanged when creating a pseudo-positive sample. Otherwise, the only difference between the pseudo-positive sample and the positive sample at the input end will be the change in positional embeddings, which we consider is insufficient to distinguish it from the positive sample.

---

> > ### Comment · Reviewer_7oUE · 2023-08-14
> >
> > Thanks for the author's response to my questions. I keep my score unchanged.

---

> > > ### Author Response · Authors · 2023-08-14
> > > **Thanks for Recognition**
> > >
> > > We appreciate your unwavering recognition of BCR!

---

### Official Review · Reviewer_qdHL · 2023-07-26

**Soundness:** 3 good
**Presentation:** 3 good
**Contribution:** 3 good
**Rating:** 6
**Confidence:** 3

**Summary:**

This paper proposes a novel turn-level dialogue evaluation framework, called Better Correlation and Robustness (BCR), which introduces pseudo-negative samples with real-valued pseudo labels to train a model that evaluates responses given a context and query. The authors also propose a new loss function that models a uniform score distribution for increased robustness, named DP loss. Through experiments, the authors demonstrate that the BCR framework can be applied to existing turn-level dialogue evaluation models, leading to improved performance in terms of correlation with human scores.

**Strengths:**

One of the main strengths of this paper is suggesting an interesting and robust method for evaluating generated responses in a conversation. The simplicity of the idea allows other researchers to easily adapt it to their own evaluation models. Furthermore, the authors present numerous experimental results demonstrating that the suggested framework outperforms the baseline model that does not incorporate the BCR framework.


**Weaknesses:**

I have some questions, so I hope to listen to the answers from the authors.

**Questions:**

- Can we explore alternative strategies to create augmented samples and soft pseudo labels? For instance, [1] considers not only the speaker but also the conversation partner of the speaker. What are other potential strategies that could be used for the algorithm?
- Is the BCR framework robust against adversarial attacks, such as removing stopwords, replacing words with synonyms, or copying one of the utterances in the context as the generated response [1, 2, 3]?
- Is the hyper-parameter beta in the DP loss sensitive? It undergoes multiple updates, so it appears crucial to determine an optimal value for improved performance.

[1] Bak, JinYeong, and Alice Oh. "Speaker Sensitive Response Evaluation Model." In Proceedings of the 58th Annual Meeting of the Association for Computational Linguistics, pp. 6376-6385. 2020.
[2] Sai, Ananya B., Mithun Das Gupta, Mitesh M. Khapra, and Mukundhan Srinivasan. "Re-Evaluating ADEM: A Deeper Look at Scoring Dialogue Responses." (2019).
[3] Sai, Ananya B., Akash Kumar Mohankumar, Siddhartha Arora, and Mitesh M. Khapra. "Improving dialog evaluation with a multi-reference adversarial dataset and large scale pretraining." Transactions of the Association for Computational Linguistics 8 (2020): 810-827.


**Limitations:**

This paper proposes an evaluation model for generated responses from an AI model, making it difficult to envision the potential negative societal impact of their work. However, I have one question related to this aspect.
Can we assess social biases, such as gender and race, using the BCR framework? My concern is that if BCR does not consider social biases, it might inadvertently reinforce bias when applied to promote the development of a dialogue system and LLM training, as the authors mentioned in the Conclusion section. Since BCR primarily focuses on aligning the overall human score of the generated response, we need to consider not only total appropriateness but also other aspects, such as context consistency, factual consistency, and social unbiasedness. This question is meant for discussion, and I hope to hear the authors' insights on this matter.

---

> ### Author Rebuttal · Authors · 2023-08-07
>
> We thank the reviewer for the positive comments as well as constructive suggestions! Below, we discuss each of the reviewer's concerns in detail.
> > **Can we explore alternative strategies to create augmented samples and soft pseudo labels? For instance, [1] considers not only the speaker but also the conversation partner of the speaker. What are other potential strategies that could be used for the algorithm?**
>
> The rationale behind applying TSR is to balance the model's evaluation capability for samples with different levels of coherence by supplementing training data with medium coherence. In addition to TSR, there may be other methods worth exploring. Here are some of our hypotheses:
> - As you mentioned, [1] considered four scenarios, including SC, SP, SS, and Random, to obtain discriminative hard negatives for training. We believe that combining our pseudo-label generation method with these scenarios could bring these samples closer to the real distribution, potentially resulting in negative samples of medium coherence (which depends on the level of coherence achieved through human evaluation of this data augmentation approach).
> - We can also consider using data from different domains. For example, we know a priori that dialogues in Dailydialog exhibit higher coherence, whereas coherence in collective discussion datasets from forums (such as Twitter) is relatively lower due to weaker targeted responses. Leveraging this prior knowledge, we can attempt to construct a training set with a more balanced coherence level.
>
> > **Is the BCR framework robust against adversarial attacks, such as removing stopwords, replacing words with synonyms, or copying one of the utterances in the context as the generated response ?**
>
> As shown in lines 252-253, we attempted to test the performance of BCR against adversarial attacks through random deletion and synonym replacement. Experimental results (Figure 7) indicate that after applying BCR, TDEMs achieved significantly higher Diff values (-0.016 -> -0.005 for GRADE and -0.010 -> -0.004 for USR). According to the definition of Formula 10, a higher Diff value indicates lesser performance degradation under adversarial attacks, thereby demonstrating that BCR is effective in countering this type of attack.
>
> We replaced responses with randomly selected sentences from the preceding context in the FED dataset and tested the average relative scores of BERT and BERT+BCR on such negative examples (average score of the model on these negative examples - average score of the model on the FED dataset). The results showed a relative score of -0.23 for BERT+BCR and a relative score of +0.15 for BERT. This indicates that the BCR framework is more effective in mitigating this type of noise and identifying such negative responses. We believe this is because treat utterances in the context as negative samples for training data construction in TSR, which theoretically enables better resistance against this type of attack.
>
> > **Is the hyper-parameter $\beta$ in the DP loss sensitive? It undergoes multiple updates, so it appears crucial to determine an optimal value for improved performance.**
>
> We observed that $\beta$ tends to reach convergence after several epochs (generally before halfway through training) across different settings (the initial value of $\beta$, etc). For example, in the experiments shown in Figure 8, among the six different values of ($s$,$l$), five groups of $\beta$ converged to 100, while another group converged to 10. A $\beta$ value of 100 appears to be a suitable choice, but for different settings (task, dataset, model), determining the optimal $\beta$ value in advance is challenging. Therefore, we recommend using DP loss to assist in finding the best $\beta$ value for a specific setting.
>
> > **Open Discussion: Bias issues in automatic dialogue evaluation.**
>
> In fact, we are currently engaged in relevant research (the impact of shortcuts on ADEM). Preliminary validation results indicate that existing TDEMs do indeed exhibit some gender and racial biases (models tend to assign lower scores to female soldiers and male nurses, for instance). Therefore, investigating how to automatically assess and mitigate social biases in generated content using automated evaluation models is a valuable avenue for future research. Given the lack of prior work in this area, we are proposing some potential research directions for discussion:
>
> - The first direction involves creating a manually annotated dataset of social biases, training models through regression tasks to automatically score the bias in sentences.
> - If constructing a comparative dataset specifically targeting social biases proves to be labor-consuming or challenging, an alternative could be to perform data augmentation during self-supervised training of ADEM (such as replacing gender and race-related terms in the context-response), in order to mitigate the impact of bias shortcuts on model scoring.
>
> We are more than willing to engage in further discussions on this matter, and any ideas would be greatly appreciated!

---

> ### Author Response · Authors · 2023-08-18
>
> Dear Reviewer qdHL,
>
> We would like to thank you again for the time you dedicated to reviewing our paper. We believe that we have addressed your concerns. Since the end of discussion period is getting close and we have not heard back from you yet, we would appreciate it if you kindly let us know of any other concerns you may have, and if we can be of any further assistance in clarifying any other issues.
>
> Thanks a lot again, and with sincerest best wishes
>
> Authors

---

### Author Rebuttal · Authors · 2023-08-10

We sincerely thank all the reviewers for their valuable feedback. We have provided detailed responses to these suggestions, and if there are still any questions that have left you confused, we eagerly look forward to providing further clarification for you.

As demonstrated by our presentation, our work provides a solid theoretical foundation for the robustness of regression tasks based on sequential relationships, and holds significant importance for advancing automatic dialogue evaluation tasks in the era of LLMs.

Looking forward to your further recognition of our work.

---

### Decision · Program_Chairs · 2023-09-21

**Decision:**

Accept (poster)

**Comment:**

In this paper, the authors address issues in turn-level dialogue evaluation models (TDEMs), namely, lower correlations between human judgments and self-supervised learning (SSL) models and nonuniform score distributions. They propose a Training Set Reconstructing (TSR) method to balance data distribution and introduce a Dynamic Penalty (DP) loss function to mitigate score distribution disparities. The proposed methods are evaluated across 17 benchmark datasets with BERT-based models, yielding promising results. One notable strength of this work is its simple yet effective approach, making it adaptable for other researchers. The motivation and problem formulation are clear, and the manuscript's organization facilitates easy comprehension. The robustness and efficacy of the proposed framework are well-supported by comprehensive experiments and evaluations.

However, several weaknesses exist and should be addressed. The novelty of the TSR technique is limited, as one reviewer pointed out that it's commonly used in dialogue systems. The paper lacks guidance on hyper-parameter tuning, and there is uncertainty surrounding certain numerical choices within the method. Comparing the proposed method with up-to-date state-of-the-art baselines are required. While the motivation for this research is sound and the proposed method offers a fresh perspective on TDEMs, it primarily focuses on a narrow scope. A broader discussion on its applicability to other tasks and its computational costs would enhance the paper.